# Uptake of BF Dye from the Aqueous Phase by CaO-g-C₃N₄ Nanosorbent: Construction, Descriptions, and Recyclability

**Ridha Ben Said** [1,2,*], **Seyfeddine Rahali** [1], **Mohamed Ali Ben Aissa** [1], **Abuzar Albadri** [3] and **Abueliz Modwi** [1]

1. Department of Chemistry, College of Science and Arts, Qassim University, Ar Rass, Saudi Arabia
2. Laboratoire de Caractérisations, Applications et Modélisations des Matériaux, Faculté des Sciences de Tunis, Université Tunis El Manar, Tunis 2092, Tunisia
3. Department of Chemistry, College of Science, Qassim University, Buraydah 52571, Saudi Arabia
* Correspondence: 141255@qu.edu.sa

**Abstract:** Removing organic dyes from contaminated wastewater resulting from industrial effluents with a cost-effective approach addresses a major global challenge. The adsorption technique onto carbon-based materials and metal oxide is one of the most effective dye removal procedures. The current work aimed to evaluate the application of calcium oxide-doped carbon nitride nanostructures (CaO-g-C₃N₄) to eliminate basic fuchsine dyes (BF) from wastewater. CaO-g-C₃N₄ nanosorbent were obtained via ultrasonication and characterized by scanning electron microscopy, X-ray diffraction, TEM, and BET. The TEM analysis reveals 2D nanosheet-like nanoparticle architectures with a high specific surface area (37.31 m²/g) for the as-fabricated CaO-g-C₃N₄ nanosorbent. The adsorption results demonstrated that the variation of the dye concentration impacted the elimination of BF by CaO-C₃N₄ while no effect of pH on the removal of BF was observed. Freundlich isotherm and Pseudo-First-order adsorption kinetics models best fitted BF adsorption onto CaO-g-C₃N₄. The highest adsorption capacity of CaO-g-C₃N₄ for BF was determined to be 813 mg·g⁻¹. The adsorption mechanism of BF is related to the π-π stacking bridging and hydrogen bond, as demonstrated by the FTIR study. CaO-g-C₃N₄ nanostructures may be easily recovered from solution and were effectively employed for BF elimination in at least four continuous cycles. The fabricated CaO-g-C₃N₄ adsorbent display excellent BF adsorption capacity and can be used as a potential sorbent in wastewater purification.

**Keywords:** calcium oxide-doped carbon nitride nanostructures; basic fuchsine; elimination mechanism; π-π stacking

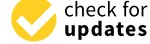



## 1. Introduction

Water pollution is one of the most important environmental hazards in the modern world, caused by wastewater discharge, insufficient treatment methods, and leakage into the natural water cycle [1,2]. Depending on the source, such as industrial plants, wastewater streams can contain excessively polluting components. Organics [3] (phenolic compounds, dyes, halogenated compounds, oils, etc.) and heavy metals (Hg, Cd, Pb, Cr, Ag, etc.) [4] are potential contaminants in wastewater, as they are biodegradable, volatile, and recycled organic compounds, suspended particles, pathogens, and parasites. Most chemical dyes are probable carcinogens [5]. Thus, before discharging wastewater, it is important to lessen or remove the presence of these potentially fatal substances.

Among these dyes, basic fuchsin BF is a triarylmethane dye that is inflammable and has antibacterial and fungicidal characteristics [6,7]. It is commonly employed as a colorant in textile and leather goods as well as in the staining of collagen and tubercle bacillus [8]. Because of its low biodegradability and its toxicity, carcinogenicity, and unsightliness [9–11], Basic Fuchsin removal from wastewater systems is a major concern that should be studied and executed as soon as possible.

In addition to the traditional biological, electrochemical, and photocatalytic oxidation and decomposition routes, physical processes (such as adsorption) are common methods that have also been developed and are used to remove organic pollutants from wastewater streams [12–14]. Even though these technologies can turn organic pollutants into non-hazardous molecules and can be used in various ways, their inability to be scaled up is a significant problem from an engineering point of view.

More specifically, the adsorption method was widely regarded as the most effective way to treat dye wastewater because of its significant adsorption capacity, low cost, good selectivity, and ease of operation [15–18]. Therefore, many researchers invest a lot of time and effort into creating new adsorbents, as well as adsorption mechanisms and treatment technology, in the hopes that they would be more useful in the treatment of dye wastewater [19–21].

Besides, graphitic carbon nitride (g-C$_3$N$_4$) nanosheet has been identified as an indispensable material for two-dimensional structures due to its graphitic-like structure and high stability under ambient circumstances [22]. It is composed of carbon and nitrogen and is most commonly employed for energy conversion and storage. Its $\pi$ conjugated polymeric metal-free semiconducting 2D structure is composed of graphitic planes composed of sp2-hybridized carbon and nitrogen [23]. Because g-C$_3$N$_4$ contains a sufficient number of edge amino and amino groups (NH/NH$_2$), it can supply several binding sites. Therefore, g-C$_3$N$_4$ is regarded as a suitable adsorbent for removing pollutants from wastewater. Nevertheless, g-C$_3$N$_4$ nanosheets capability to adsorb is limited by its small surface area and few functional groups [24].

Therefore, the development of g-C$_3$N$_4$-containing compounds with higher photonic efficiency, such as TiO$_2$ and ZnO, piqued the curiosity of a vast number of researchers [25,26]. This was accomplished by combining g-C$_3$N$_4$ with another semiconductor and decorating g-C$_3$N$_4$ with noble metals [27–31]. Construction of heterojunctions comprised of g-C$_3$N$_4$ mixed with another type of compound, such as CaO nanomaterials, and preparation of a Ca-O doped with g-C$_3$N$_4$ with an improved surface texture by selecting the optimal preparation method are the most beneficial means of enhancing the adsorption properties of g-C$_3$N$_4$.

In the current study, a mesoporous CaO@g-C$_3$N$_4$ nanocomposite was successfully produced using a simple sonochemical process and evaluated as a promising adsorbent material for adsorbing the basic fuchsin dye from a contaminated aqueous phase. The physicochemical relationship between characterizations and measurements of equilibrium and kinetics was studied. Adsorption isotherm data were also modeled, and the adsorption performance of CaO@g-C$_3$N$_4$ nanocomposite for basic fuchsin was investigated.

## 2. Experimental

### 2.1. Chemicals

Sodium hydroxide (NaOH, ≥99%), sodium chloride (NaCl, ≥99%), basic fuchsin (BF, ≥85%), urea (CH$_4$N$_2$O, ≥98%) and calcium carbonate (CaCO$_3$, ≥99%) purchased from Merck Company were used without further purification. The required dyes concentrations (25, 50, 100, 150, 200, and 300 ppm) were obtained by diluting BF stock solution (500 ppm).

### 2.2. CaO-g-C$_3$N$_4$ Nanosorbent Fabrication

The nanosheets of g-C$_3$N$_4$ were produced through the thermal breakdown of urea. In a typical technique, 0.075 moles of a carbamide compound were placed in a covered pot and tempered with a heating rate of 10 °C/min at 723 K for 120 min. The produced yellow raw g-C$_3$N$_4$ was then cooled, pulverized, and stored in a dark container. Thermally decomposing carbonate salts created calcium oxide (CaO) nanoparticles. Two grams of calcium carbonate salts were weighed, placed in a crucible, and annealed at 1073 K for one hour. CaO-g-C$_3$N$_4$ nanoparticles were produced using a step-by-step ultrasonication technique aided by an organic solvent (ethanol). In 125 mL of ethanol, 2.76 mg of g-C$_3$N$_4$ was sonicated for 15 min. CaO nanoparticles were added to the g-C$_3$N$_4$ ethanolic solution

along with an additional 45 min of sonication. The yellowish solution generated was evaporated at 368 K for 1440 min. CaO-g-C$_3$N$_4$ nanosorbent was ultimately tempered at 453 K for 60 min.

### 2.3. CaO-g-C$_3$N$_4$ Nanosorbent Characterizations

The nanosorbent CaO-g-C$_3$N$_4$ was studied using a variety of analytical and spectroscopic techniques. Energy dispersive X-ray (EDX) spectroscopy was used to calculate the elemental composition of the CaO-g-C$_3$N$_4$ nanosorbent. The transmission electron microscope (Tecnai G20-USA) was used to make morphological observations, and the stimulating voltage was set at 200 kV. X-ray diffraction (XRD) was used to analyze the phase structure using a Bruker D8 Advance diffractometer Cu-K ($\lambda$ = 1.540) radiation source. An ASAP 2020 device was used to evaluate the accurate analysis of the surface area. Before and after the BF dye elimination, Fourier transformed infrared (FTIR) spectra were recorded using a Nicolet 5700 spectrometer equipped with a KBr pellet.

### 2.4. BF Dye Removal Experiments

By mixing 25 mL of BF dye solution with 10 mg of CaO-g-C$_3$N$_4$ nanosorbent at varying starting concentrations (5–300 mg/L), batch removal tests of BF dye were conducted. In order to attain equilibrium, the set mixture was stirred for 24 h at 400 rpm. After centrifugation, a clear solution was produced. The dye volumes and beginning concentrations were 100 mL and 250 ppm, respectively, for the kinetic experiment, and the CaO-g-C$_3$N$_4$ nanosorbent mass was 40 mg. The test was conducted in the dark with magnetic stirring. Later, 10 mL of the suspension was withdrawn and centrifuged for 10 min to measure the remaining concentration of BF dye.

Using a spectrophotometer, the concentration of dye was determined, and equilibrium dye capacity ($q_e$) was calculated using the following equation:

$$q_e = \frac{C_0 - C_e}{m} \cdot v \tag{1}$$

where $q_t$ (mg·g$^{-1}$) is the quantity of dye removed by a unit mass of nanosorbent $m$ (g) at a specified interval of time (min), $V$ is the volume of the solution (L), $C_0$ is the initial dye concentration, and $C_t$ is the concentration at time $t$ (mg L$^{-1}$). A similar calculation was used to compute the amount adsorbed at equilibrium, $q_t$:

$$q_t = \frac{C_0 - C_t}{m} \cdot v \tag{2}$$

The influence of the dye's elimination on the pH of the aqueous media was investigated by setting the initial pH value of dye solution from 3 to 11 pH range by using either NaOH (0.1 mole·L$^{-1}$) or HCl (0.1 mole·L$^{-1}$). The pH of zero-point charge (pHzpc) for CaO-g-C$_3$N$_4$ nanosorbent was evaluated by the salt addition approach. A fixed amount of CaO-g-C$_3$N$_4$ nanosorbent (20 mg) was added in each flask containing 20 mL NaCl solution (0.01 M) with pH initial (pHi) values raised from 2 to 12 (by adjustments using 0.1 M NaOH or HCl solutions). The mixture was stirred for one hour, and the final pH (pH$_f$) was calculated after eliminating CaO-g-C$_3$N$_4$ by filtration. For the reusability test, the CaO-g-C$_3$N$_4$ nanosorbent used after the adsorption experiments was recovered by filtration and then calcined at 773 K. for one hour. After that, the recovered CaO-g-C$_3$N$_4$ nanosorbent was reused for further adsorption tests.

## 3. Results and Discussion

### 3.1. CaO-g-C$_3$N$_4$ Nanosorbent Characterizations

The scanning elemental mapping analysis for Ca, N, O, and C in the CaO-g-C$_3$N$_4$ nanosorbent aggregates (Figure 1a–e) indicates an overall homogeneous dispersion, as shown in Figure 1b–e. On the elemental maps, a brighter zone implies a higher elemental ratio. The CaO-g-C$_3$N$_4$ nanocomposite has created a homogenous distribution, according

to this observation. An image taken using EDX identifies the individual components that are present in CaO-g-C$_3$N$_4$ nanosorbent material. As a result, it is clear from the findings of the EDX performed on CaO-g-C$_3$N$_4$ nanosorbent that the surface is composed of carbon (C), nitrogen (N), calcium (Ca), and oxygen (O). These findings are because the results depict bands corresponding to each component (Figure 1a–f).

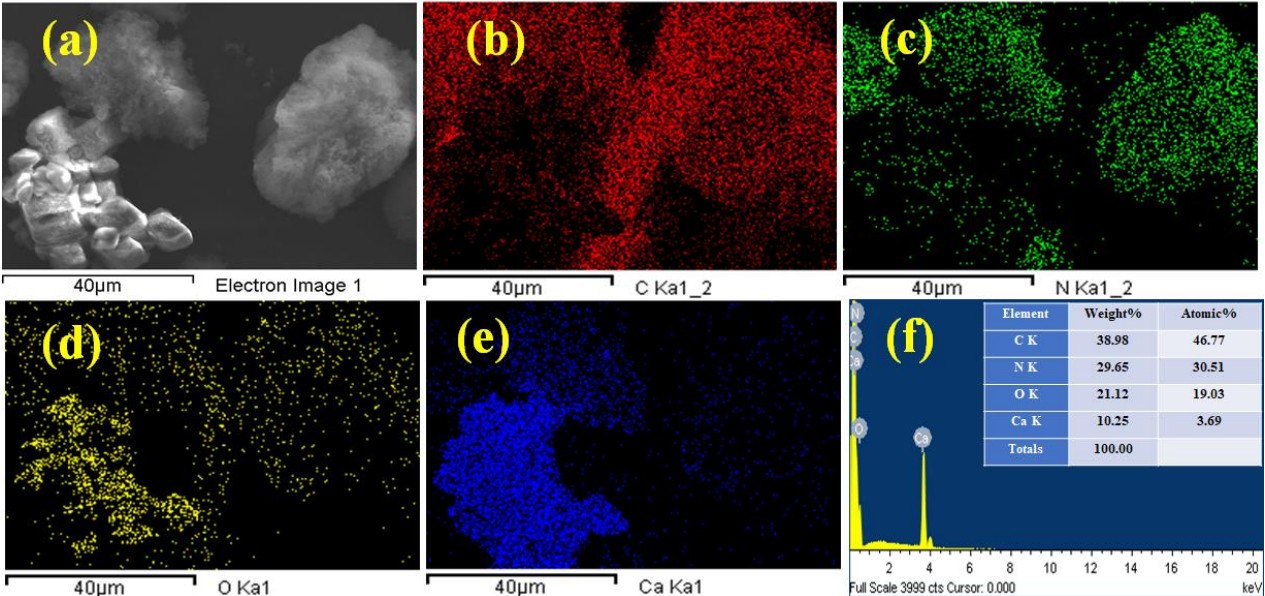

**Figure 1.** (**a–e**) Elemental mapping distribution and (**f**) EDX graph of CaO-g-C$_3$N$_4$ nanosorbent.

The TEM micrograph was utilized to investigate the textural qualities of the fabricated CaO-g-C$_3$N$_4$ nanosorbent. The TEM photographs of the CaO, g-C$_3$N$_4$, and CaO-g-C$_3$N$_4$ nanostructures are depicted in Figure 2. The TEM image of CaO (Figure 2a) presents like-sheets shape nanoparticles. On the other hand, the TEM image of g-C$_3$N$_4$ (Figure 2b) displays layers with soft surface sheets as a typical graphitic nanostructure [32]. Furthermore, the images with different magnifications obtained from the TEM of the CaO-g-C$_3$N$_4$ demonstrated that the morphology had an apparently random appearance, as given in (Figure 2a–d). The CaO-g-C$_3$N$_4$ nanosorbent exhibits characteristic 2D nanosheet-like nanoparticle architectures, as seen in (Figure 2a–d). The average particle size of the CaO nanoparticles integrated into the CaO-g-C$_3$N$_4$ nanosorbent is around 20–60 nm. CaO are observed to be well disseminated on the g-C$_3$N$_4$ surface, forming an abundance of self-active sites on the nanosorbent surface.

Figure 3 depicts the typical peaks at 12.92° and 27.64° for g-C$_3$N$_4$ in the XRD pattern. The first peak corresponds to the in-plane packing of tris-triazine units with a d-spacing of 0.685 nm, which agrees with the distance between holes in the nitride pores. However, the great peak at 27.64° is associated with C–N aromatic stacking units separated by 0.322 nm, which corresponds to the 002 interlayer layering plane of the connected aromatic system [33]. Alternatively, the peaks 32.14, 37.25, 53.77, 64.00, and 67.30° correspond to the (110), (200), (202), (311), and (222) planes of the cubic phase of CaO (XRD file JCPDS 77-2376) [34]. Besides the CaO peaks, Ca (OH)$_2$ and CaCO$_3$ peaks are seen at 18,00°, 29,32°, 47.39°, and 48.40°, respectively. The presence of calcite (CaCO$_3$) and hydroxide peaks indicates incomplete pyrolysis of the precursor and fast carbonation and hydrolysis of CaO by ambient CO$_2$ and water vapor. Literature indicates that CaO nanoparticles have a strong potential for capturing greenhouse gas CO$_2$ [35]. Finally, the XRD pattern obtained from the fabricated indicates the presence of the g-C$_3$N$_4$ and CaO peaks (Figure 3), implying the construction of the target nanosorbent.

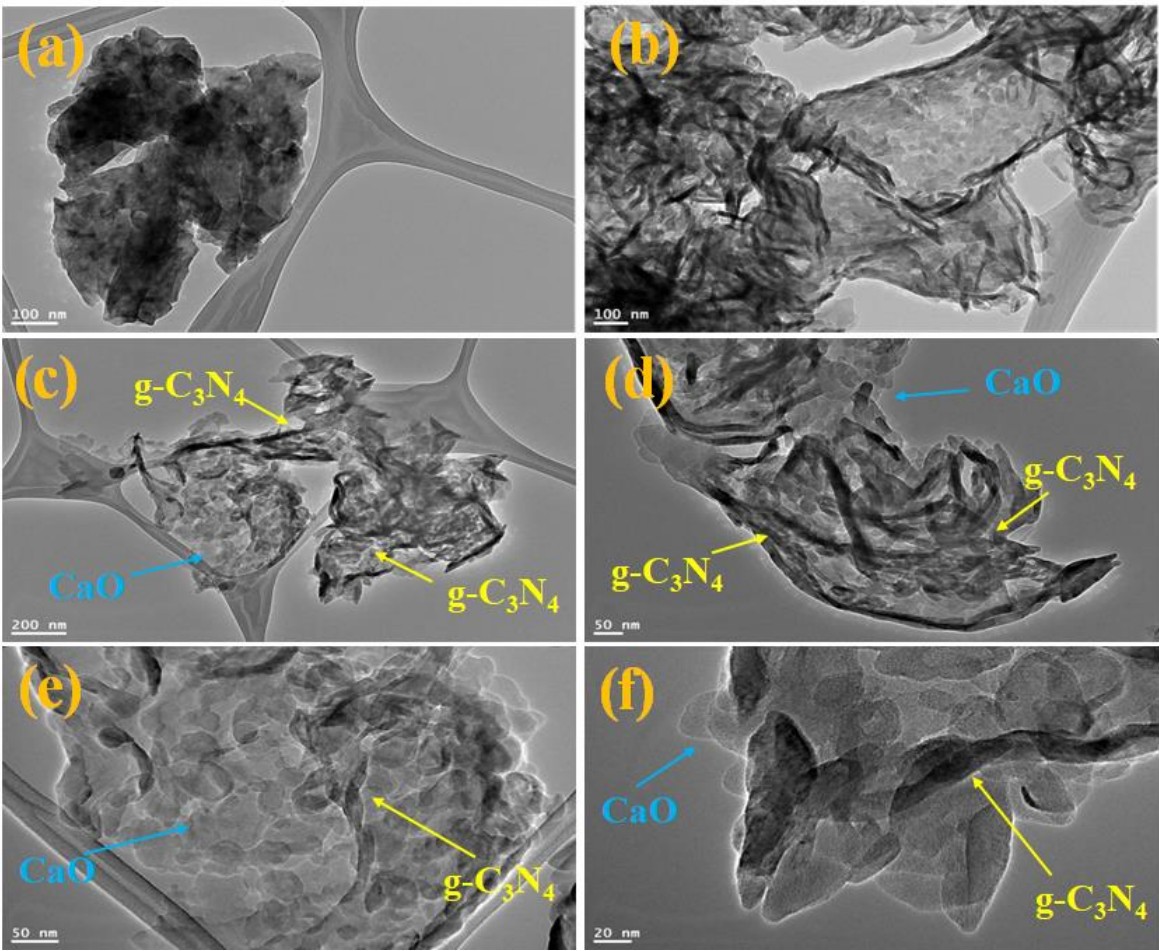

**Figure 2.** TEM images of (**a**) CaO, (**b**) g-C₃N₄, and (**c**–**f**) CaO-g-C₃N₄ nanosorbent with different magnifications.

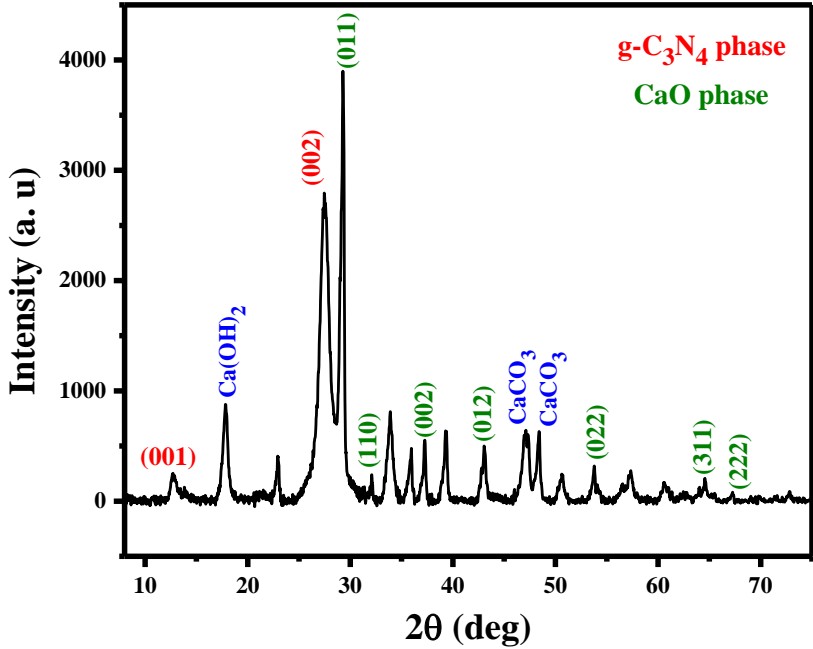

**Figure 3.** XRD patterns of CaO-g-C₃N₄ nanosorbent.

Nanomaterials utilized as adsorbents are profoundly influenced by their particular surface area and porous structure, which can provide additional adsorption and reactive sites. The N2 absorption-desorption isotherms of CaO-g-$C_3N_4$ nanosorbent, which may be categorized as type IV according to the IUPAC system, were determined [36]. Figure 4a,b displays the BET surface isotherms and pore size distribution of the CaO-g-$C_3N_4$ nanosorbent as manufactured. According to the results, the CaO-g-$C_3N_4$ nanosorbent absorption-desorption graphs fit isotherm type IV and the hysteresis loop ($H_2$) for relative pressures between 0.0 and 1.0. This result confirmed the mesoporous nature of the CaO-g-$C_3N_4$ nanosorbent [37,38]. Due to the presence of several active sites on the surface, the CaO-g-$C_3N_4$ nanosorbent increased surface area, demonstrated by a higher specific surface area (37.31 $m^2$/g) with a pore volume of 0.136 cc/g, will improve the sorption capacity [39].

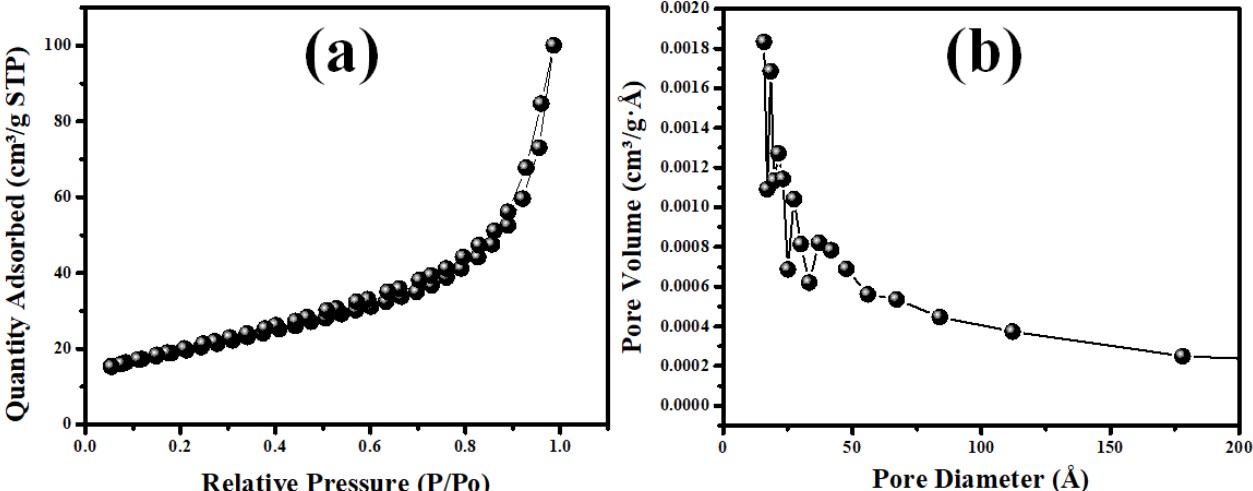

**Figure 4.** (**a**) BET surface isotherms and (**b**) pore sizes distribution of the CaO-g-$C_3N_4$ nanosorbent.

The chemical condition of the elements on the surface of the CaO@g-$C_3N_4$ nanostructure was determined by XPS analysis; see Figure 5a–d. CaO exists because the Ca 2p peaks at 349 and 352.6 eV correspond to the divalent oxidation states of calcium oxygen molecules (Ca-2p3/2 and Ca-2p1/2) [40,41]. Ca-O and hydroxyl groups in water molecules correspond to the three O 1s peaks detected at 532.6, 533.7, and 534.4 eV which correspond to the lattice oxygen of the layer-structured Ca-O, and adsorbed $H_2O$ or surface hydroxyl oxygen, respectively (Figure 5b) [42]. As shown in Figure 4c, carbon in the C-C and N-C=N states is attributed with two distinct contributions at 285.8 and 288.2 eV, respectively. According to the XPS analysis, only Ca, O, C, and N are present. The peaks at 398.8 and 400.3 eV (Figure 5d) for N 1s which indicate, respectively, the sp2 hybridized carbon–nitrogen bonding in (C–N) and N-O of the CN and the binding energy of the N atoms in C-N-C [43]. The absence of other impurity peaks supports the results of the XRD and EDS studies.

### 3.2. BF Removal onto CaO-g-$C_3N_4$ Nanosorbent

### 3.2.1. Impact of Variation pH on BF Removal by CaO-g-$C_3N_4$ Nanosorbent

The solution's pH controls the adsorbent's sorption affinity by adjusting the surface charge and the ionizing strength of the adsorbent [44]. Adsorption experiments of CaO-g-$C_3N_4$ nanosorbent were conducted under various initial pH values in order to demonstrate the impact of pH value on the adsorption of BF dyes (from 3 to 11). Figure 6a illustrates the influence of pH on BF uptake. It was discovered that BF dyes may be stably adsorbed without observable alterations. The zero-point charge experiment was performed to explain the acquired results. The pHzpc of CaO-g-$C_3N_4$ nanosorbent is determined to be 10.6, as shown in Figure 6b. At lower pH, the surface of the CaO-g-$C_3N_4$ nanosorbent is positively charged and becomes negative at pH greater than pHpzc (=10.6). The pH studies showed that the adsorption is pH-independent, indicating that the electrostatic interactions do not

control the adsorption mechanism. The adsorption was likely due to the formation of H bonding between -OH and -$NH_2$ onto the CaO-g-$C_3N_4$ surface with BF dye molecules [45].

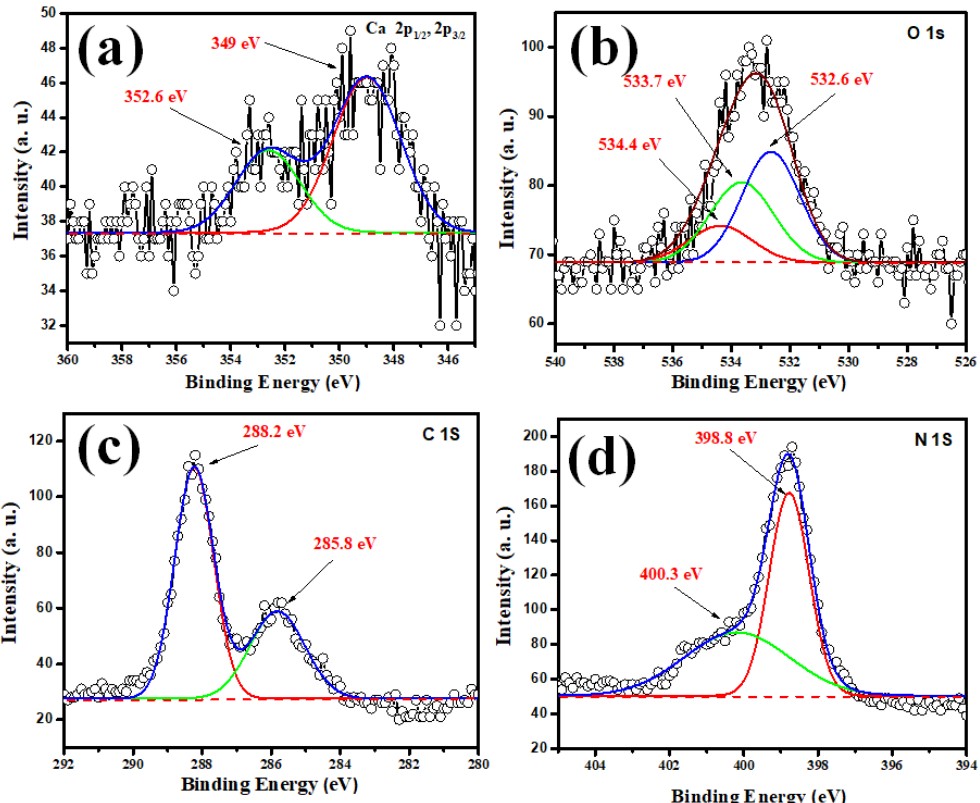

**Figure 5.** XPS spectra of (**a**) Ca-2p; (**b**) O 1s; (**c**) C 1s and (**d**) N 1s for CaO@g-$C_3N_4$ nanosorbent.

### 3.2.2. Influence of the Initial BF Dye Concentration and Doping

The influence of the initial BF dye concentration in the range of 5–300 mg/L on the adsorption efficiency of CaO-g-$C_3N_4$ nanosorbent was scrutinized under the following operating conditions: contact time 1440 min, room temperature, pH 7, 400 rpm stirring speed, and a CaO-g-$C_3N_4$ sorbent dose of 10 mg. As shown in Figure 6c, increasing the initial BF concentration from 5 to 300 mg/L improved the adsorption capacity significantly from 60.61 to 738.08 mg/g. These results indicate that BF molecules in the reaction medium interact more strongly with the top layer of the CaO-$C_3N_4$ sorbent particles at lower concentrations due to a large amount of vacant active sites. Conversely, the ratio of accessible sites for BF molecules declines with a further rise in the concentration attributed to saturation. To compare the BF dye adsorption capacities of g-$C_3N_4$ and CaO-g-$C_3N_4$, a series of adsorption experiments were conducted at different initial BF concentrations and a pH value of 7. The obtained results are also shown in Figure 6c. It is interesting to remark that CaO-g-$C_3N_4$ exhibits higher BF adsorption capacity than the respective capacities of pure g-$C_3N_4$ for the different initial BF concentrations. This result demonstrates that g-$C_3N_4$ nanosheets have the capability to adsorb is limited by its small surface area and few functional groups. The doping of g-$C_3N_4$ by CaO enhances the adsorption properties of g-$C_3N_4$ by improving surface texture.

### 3.2.3. Adsorption Equilibrium of BF Dyes onto CaO-$C_3N_4$ Nanosorbent

The greatest amount of BF absorbed by CaO-g-$C_3N_4$ nanosorbent is a crucial characteristic for assessing the high adsorption capacity exhibited. To estimate the absorption capacity of CaO-g-$C_3N_4$ nanosorbent, two adsorption isotherm models (Freundlich and Langmuir) were utilized to assess the adsorption data, as depicted in Figure 7a,b. Table 1 contains the formulas corresponding to each isotherm model and the derived parameters.

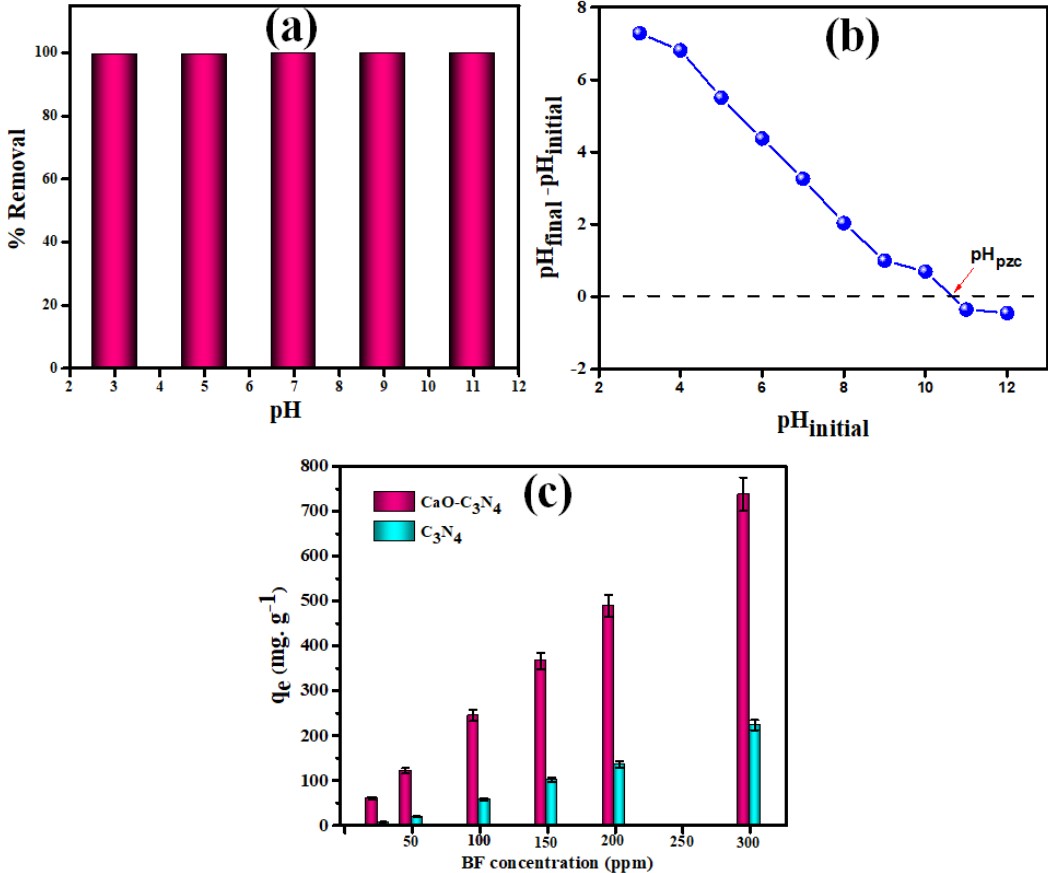

**Figure 6.** (**a**) Effect of pH on % removal of BF, (**b**) plot for the determination of $pH_{pzc}$ for CaO-g-$C_3N_4$ nanosorbent, and (**c**) influence of dye concentration onto the adsorption capacity of CaO-g-$C_3N_4$ and g-$C_3N_4$ nanosorbent.

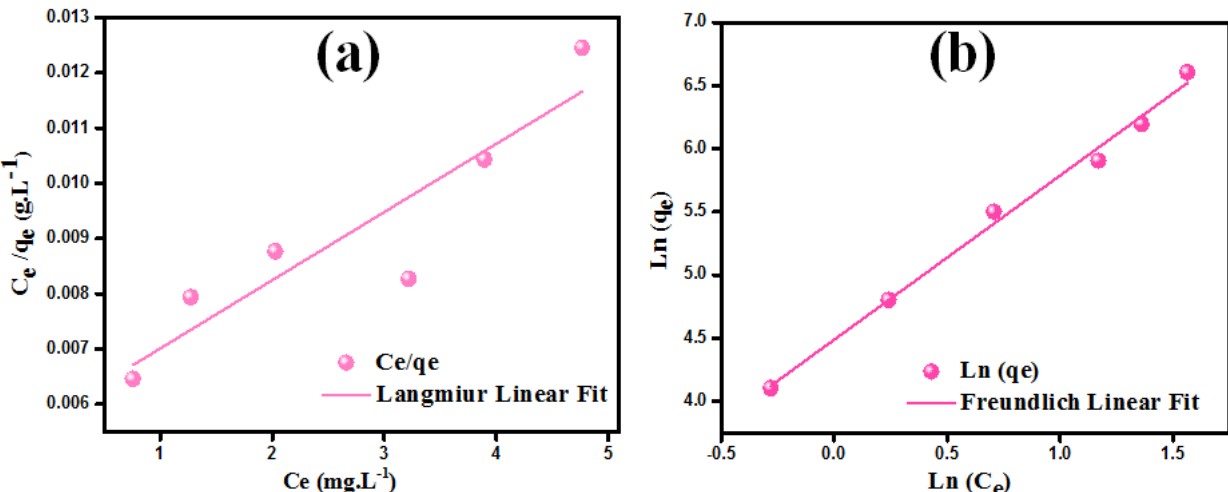

**Figure 7.** Experimental and fitted adsorption data using the Freundlich (**a**) and Langmuir (**b**) models.

As can be noted from the isotherm graphs and the experimental data for BF adsorption over CaO-$C_3N_4$ nanosorbent, the Freundlich adsorption isotherm has the highest $R^2 = 0.996$. These findings indicate that the Freundlich adsorption isotherm curve is more accurate to fit the experimental data.

**Table 1.** Used equilibrium isotherm models for the adsorption of BF onto CaO-g-C$_3$N$_4$ nanosorbent.

| Equilibrium Models | Linear Form | Parameter | Value |
|---|---|---|---|
| Langmuir [46] | $\frac{C_e}{q_e} = \frac{1}{q_m K_L} + \frac{C_e}{q_m}$ | $q_m$ (mg/g) | 813.0 |
| | | $K_L$ (mg/g) | 0.212 |
| | | $R_L$ (L/mg) | 0.0015 |
| | | $R^2$ | 0.842 |
| Freundlich [47] | $lnq_e = lnK_F + \frac{1}{n}lnC_e$ | n | 0.97 |
| | | $K_F$ (L/mg) | 88.89 |
| | | $R^2$ | 0.996 |

The greatest sorption capacity of CaO-g-C$_3$N$_4$ nanosorbent for BF dyes is found to be 813 mg·g$^{-1}$, as given in Table 1.

3.2.4. BF Contact Time and Adsorption Kinetic Studies

Figure 8a shows the relationship between contact time and BF adsorption on the surface of the CaO-C$_3$N$_4$ nanosorbent. Adsorption capacity is seen to increase with longer contact times, reaching equilibrium after 25.9 min. Beyond this equilibrium threshold, the adsorption capacity and the amount of BF adsorbed are in dynamic equilibrium. BF molecules were rapidly adsorbed by CaO-C$_3$N$_4$ nanosorbent for the first 25.9 min, after which the adsorption rate declined until it reached its maximum value at around 1440 min. The initial adsorption rates are relatively high due to the abundance of active sites on the surface of the CaO-g-C$_3$N$_4$. After attaining equilibrium, the active site concentration falls, and dye adsorption does not occur.

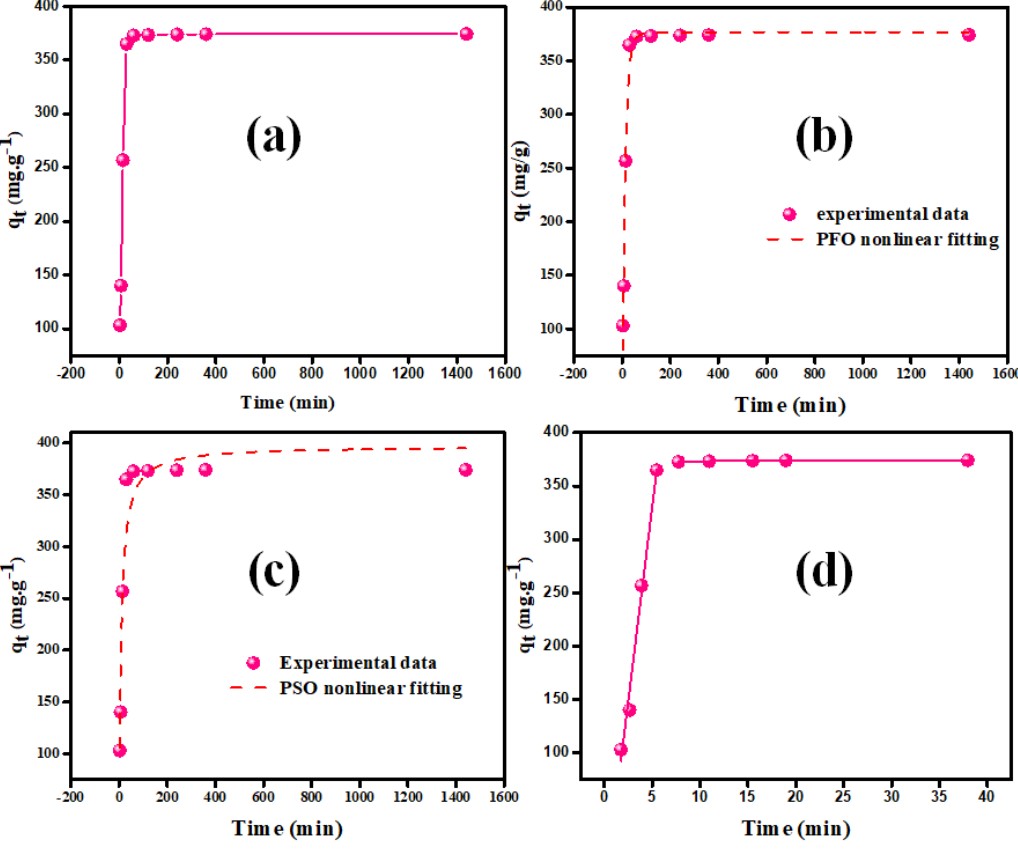

**Figure 8.** (**a**) Equilibrium time study (**b**) PFO, (**c**) PSO and (**d**) Intra-particle diffusion graphs for the uptake of BF onto CaO-g-C$_3$N$_4$.

The adsorption kinetics measures the rate of solute adsorption at the solid–liquid interface and gives essential information on the equilibrium period for the design and management of an adsorption process [48]. As shown in Figure 8b,c, the BF adsorption kinetics by the $CaO$-g-$C_3N_4$ nanosorbent was investigated using pseudo-first-order (PFO) and pseudo-second-order (PSO) kinetic models (b and c). The form of the relevant nonlinear equations is shown in Table 2.

**Table 2.** Kinetics models for BF adsorption onto $CaO$-g-$C_3N_4$ nanosorbent.

| Kinetics Models | Kinetic Equations | Parameter | Value |
|---|---|---|---|
| PFO [49] | $q_t = q_e \left(1 - e^{-1k_1 t}\right)$ | $Q_m$ (exp) (mg/g) | 375.69 |
| | | $Q_e$ (mg/g) | 376.49 |
| | | $K_1$ (min$^{-1}$) | 0.080 |
| | | $R^2$ | 0.984 |
| PSO [49] | $q_t = \dfrac{t \, k_2 q_e^2}{k_2 q_e t + 1}$ | $Q_e$ (cal) (mg/g) | 397.13 |
| | | $K_2$ (g/mg·min) | 0.0003 |
| | | $R^2$ | 0.930 |
| Intra-particle Diffusion [50] | $q_t = k_{dif}\sqrt{t} + C$ | $K_{dif1}$ (mg. min$^{1/2}$/g) | 72.78 |
| | | $C_1$ | 33.53 |
| | | $R^2$ | 0.987 |
| | | $K_{dif2}$ (mg·min$^{1/2}$/g) | 0.04 |
| | | $C_2$ | 372.83 |
| | | $R^2$ | 0.643 |

The computed model parameters under the experimental conditions tested are summarized in Table 2. The PFO model might adequately describe the experimental adsorption kinetics data. It claims that the ratio of the square of the number of accessible sites to the rate of adsorption site occupancy. The form formula for the PFO nonlinear linear model is shown in Table 2. Using the computed model parameters in Table 2, the extraordinarily high $R^2$ value of 0.984 is determined. Compared to the PSO, the PFO equation provides a perfect fit, as shown by the findings. For the tested BF concentrations, there is only a small difference between the experimental $Q_{max}$ values and the model-estimated $Q_{max}$ values. As a result, the PFO model's best fit implies that the kinetic adsorption may be mathematically described using the concentration of BF in solution [44,51].

Through the intra-particle diffusion/transport mechanism, the BF elimination may be transferred from the bulk of the solution to the solid phase of the $CaO$-g-$C_3N_4$ nanosorbent. In some circumstances, the step of the adsorption process known as intra-particular diffusion is restrictive. The diffusion pattern developed by Weber and Morris supports the notion of intra-particulate diffusion [52,53]. As $q_t$ and $t^{1/2}$ are compared linearly, the removal of BF onto the $CaO$-$C_3N_4$ surface demonstrates the efficacy of the intra-particle diffusion kinetic pattern. In addition, the intra-particle mode of diffusion is characterized by the regression coefficient ($R^2 = 0.987$). The diameter of the boundary layer is represented by parameter C's value. The higher percentages of the constants in Table 2 demonstrate the solution boundary layer's strong influence on the removal of BF dyes [52,53]. It can be seen that the first stage of sorption has a larger rate than the second phase, which is shown by $k_{dif1} > k_{dif2}$ (Table 2). The high value of the rate produced by the first step may be explained by the movement of the dye mover through the solution and onto the surface of the outer $CaO$-g-$C_3N_4$ that is generated by the boundary layer. Comparatively, the subsequent phase describes the last equilibrium step, when intra-particle diffusion begins to diminish due to the solute's modest concentration gradient and the restricted number of holes and pores available for diffusion [54].

### 3.3. Regeneration and Reusability Study

By removing BF from the surface of the nanomaterial, the reusability and regeneration of $CaO$-g-$C_3N_4$ sorbent were investigated. Following the adsorption experiment, the BF

was removed from the CaO-g-C$_3$N$_4$ by heating it in an oven at 773 K for a one hour. The recovered CaO-g-C$_3$N$_4$ was then reapplied to the BF elimination process. CaO-g-C$_3$N$_4$ has been used efficiently for the removal of BF for at least four continuous cycles, as shown by the reusability results (Figure 9a). As shown, there was no obvious decrease in the elimination effectiveness during four adsorption–desorption cycles, and only 4%, 7%, and 9% of the adsorption capacity for BF declined at the second, third, and fourth cycles, respectively.

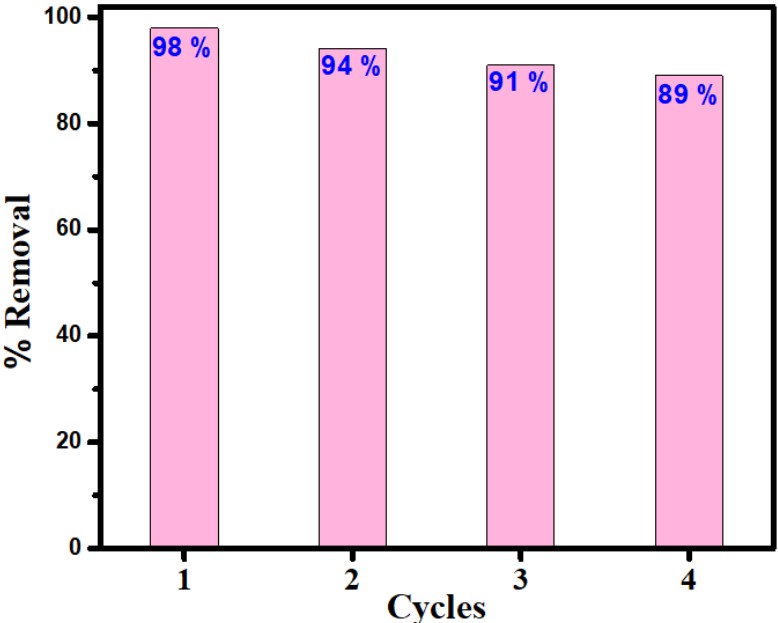

**Figure 9.** Reusability effectiveness of CaO-g-C$_3$N$_4$.

*3.4. Comparison Study*

As shown in Table 3, the calculated adsorption capacity of CaO-g-C$_3$N$_4$ for BF using the Langmuir isotherm model is 813.00 mg·g$^{-1}$. It is to one's advantage to evaluate the CaO-g-C$_3$N$_4$ adsorption capacity in relation to the diverse sorbents that can be utilized for BF elimination. Table 3 shows the various sorbents with high adsorption capacities for BF removal. Compared to previously reported sorbents like MgO and modified activated carbons, CaO-g-C$_3$N$_4$ has a higher capacity for adsorption. This finding confirmed that CaO-g-C$_3$N$_4$ is an efficient BF dye adsorbent.

**Table 3.** Observation of adsorption capacities of the CaO-g-C$_3$N$_4$ using various nanomaterial adsorbents.

| Adsorbents | $q_e$ (mg g$^{-1}$) | Best pH | BET Surface Area (m$^2$/g) | References |
| --- | --- | --- | --- | --- |
| Fe-MgO/kaolinite | 10.36 | 9.0 | - | [55] |
| YZnO nanoparticles | 75.53 | 11 | 20.26 | [56] |
| Al/MCM-41 | 54.44 | 3–9 | 997 | [57] |
| Euryale ferox Salisbury seed shell | 19.48 | 6.0 | - | [58] |
| ESM | 47.85 | 6.0 | 11.56 | [59] |
| Fe/ZSM-5 | 251.87 | 5.0 | 399 | [60] |
| Modified activated carbons | 238.10 | 8.5 | 613 | [61] |
| MgO | 493.90 | 11 | 12.22 | [62] |
| MgOg-C$_3$N$_4$ | 1250 | 7.0 | 84.2 | [63] |
| CaO-g-C$_3$N$_4$ | 813.00 | Independent | 37.31 | Current study |

*3.5. Adsorption Mechanism*

The adsorption mechanism of BF dyes by nanosorbent has been elucidated using FTIR analysis. Figure 10a depicts the FTIR spectra of nanosorbent prior to and following BF adsorption. CaO-g-C$_3$N$_4$ spectrum reveals a number of distinguishable bands: the

bandwidth between 3000 and 3600 cm$^{-1}$ corresponds to the stretching vibration modes of O–H and NH. The absorption bands at 1242, 1326, and 1412 cm$^{-1}$ are associated with aromatic C–N stretching, while the absorption bands at 1578 and 1640 cm$^{-1}$ are associated with C≡N stretching. The band at 884 cm$^{-1}$ corresponds to the triazine ring mode, a frequent carbon nitride mode. The characteristic band located at 805 cm$^{-1}$ is assigned to Ca-O vibration mode [64]. After adsorption, as can be observed in Figure 10a, many typical BF bands form and move around with respect to the free molecules, suggesting that CaO-g-C$_3$N$_4$ and BF molecules may interact. Also, following the adsorption of BF dyes, several vibration bonds of CaO-C$_3$N$_4$, such as aromatic C–N stretching and triazine ring modes, have shifted position. This study demonstrated that delocalized electron systems of C$_3$N$_3$ and functional groups of CaO-g-C$_3$N$_4$ were responsible for the adsorption of BF molecules. In addition, the O–H and NH stretching vibration modes were shifted, demonstrating the interaction of BF molecules with CaO-g-C$_3$N$_4$ nanosorbent via hydrogen bonds. Lie et al. demonstrate that sorbents containing pyrazine and imine groups are beneficial to the formation of π-π stacking and hydrogen bonds interactions with organics dyes [65]. Also, the examination of the pH's effect indicates that the electrostatic attraction could not dominate (control) the adsorption mechanism of BF onto the CaO-g-C$_3$N$_4$ nanosorbent. The suggested BF adsorption mechanism (Figure 10b) onto the CaO-g-C$_3$N$_4$ involves hydrogen bonds and the π-π stacking bridging [66].

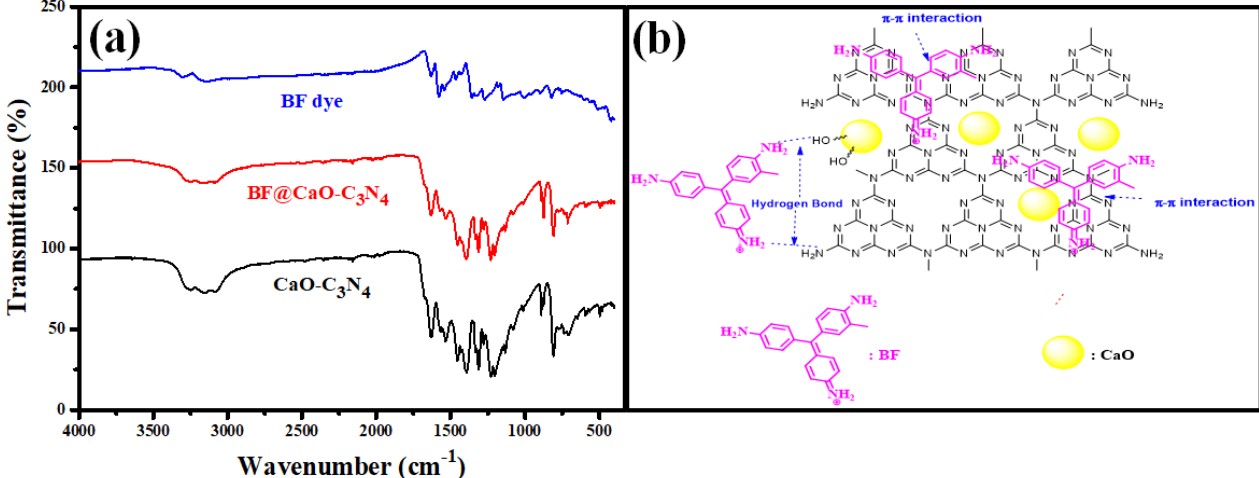

**Figure 10.** (**a**) FTIR spectra of BF, CaO-g-C$_3$N$_4$ and CaO-g-C$_3$N$_4$ @BF and (**b**) Possible adsorption mechanism of BF dyes onto CaO-g-C$_3$N$_4$.

## 4. Conclusions

Mesoporous CaO-g-C$_3$N$_4$ nanosorbent was created using the ultrasonication technique, and it was subsequently employed as an adsorbent to remove BF dyes from wastewater. CaO-g-C$_3$N$_4$ nanosorbent removal efficiency was studied by adjusting pH, contact time, and BF concentration. The greatest adsorption capacity observed was 813 mg·g$^{-1}$, indicating that the reported data demonstrated outstanding elimination effectiveness toward BF dye. The BF elimination by CaO-g-C$_3$N$_4$ nanosorbent was evaluated employed different adsorption and kinetic models, and the best-fitting was committed by the Freundlich adsorption isotherm and PFO kinetics models. The suggested BF adsorption mechanism onto the CaO-g-C$_3$N$_4$ involves hydrogen bonds and the π-π stacking bridging. CaO-g-C$_3$N$_4$ nanostructures may be easily recovered from solution and were effectively employed for BF elimination in at least four continuous cycles.

**Author Contributions:** R.B.S.: Conceptualization, Methodology, Formal analysis, Investigation, Visualization. S.R.: Methodology, Formal analysis, Investigation. M.A.B.A.: Methodology, Formal analysis, Investigation. A.A.: Conceptualization, Methodology, Formal analysis, Investigation. A.M.: Conceptualization, Methodology, Formal analysis, Investigation, Visualization. All authors have read and agreed to the published version of the manuscript.

**Funding:** The authors extend their appreciation to the Deputyship for Research & Innovation, Ministry of Education, Saudi Arabia for funding this research work through the project number (QU-IF-05-01-28461).

**Acknowledgments:** The authors extend their appreciation to the Deputyship for Research & Innovation, Ministry of Education, Saudi Arabia for funding this research work through the project number (QU-IF-05-01-28461). The authors give thanks to Qassim University for technical support.

**Conflicts of Interest:** The authors declare no conflict of interest.

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
