# Peer review of "Uptake of BF Dye from the Aqueous Phase by CaO-g-C3N4 Nanosorbent: Construction, Descriptions, and Recyclability"

_inorganics, doi:10.3390/inorganics11010044_

Round 1

Reviewer 1 Report

The following issues must be addressed:

1.       The Abstract should contain more experimental results;

2.       Introduction part must be improved to outline what is new and innovative in this work compared with other similar papers;

3.       More information’s about characterization parameters must be provided;

4.       CaO and gC3N4 TEM images should be included;

5.       Explain if you observe a preferential crystal grow.

6.       Haw, you quantified the vacant active sites?

7.       Is not clear why after 25.9 min there is a such a rapid decrease of the adsorption rate. Please explain in more details.

8.       Conclusion part must be improved to include more relevant results.

Author Response

  1. The Abstract should contain more experimental results;

More experimental results had been added in the abstract.

  1. Introduction part must be improved to outline what is new and innovative in this work compared with other similar papers;

Reply:

The manuscript rewritten and improved as well as valuable more information’s was added as per-reviewers request.

  1. More information’s about characterization parameters must be provided;

Reply:

More information’s and new characterizations results such as XPS, pore distributions and TEM images of CaO nanomaterials and pure g-C3N4 nanosheets had been added and discussed as per the reviewer's request.

  1. CaO and gC3N4 TEM images should be included;

Reply:

      The TEM images were replotted and included CaO nanoparticles and pure g-C3N4 in Fig 2a and b as per the reviewer's request.

  1. Explain if you observe a preferential crystal grow.

Reply:

The crystal grow has not been observed.

  1. Haw, you quantified the vacant active sites?

Reply:

The estimation of the vacant active sides was indicated by surface area, pore size and porosity as well as particles size, addition of extend of the adsorb quantity.

  1. Is not clear why after 25.9 min there is a such a rapid decrease of the adsorption rate. Please explain in more details.

More details were added.

"The initial adsorption rates are relatively high due to the abundance of active sites on the surface of the CaO-g-C3N4. After attaining equilibrium, the active site concentration falls, and dye adsorption does not occur."

  1. Conclusion part must be improved to include more relevant results.

Reply:

The conclusion part was improved.

Reviewer 2 Report

The manuscript on the topic Uptake of basic fuchsine dye from the aqueous phase by CaO-g-C3N4 nanosorbent: construction, descriptions, and  recyclability is a good attempt made by the authors. However, the manuscript is weak and needs major revision for reconsideration.

Comments are provided below.

1. Title of the script is not convincing. Need to be revised.

2. The author needs to confirm and clarify the novelty of this research.

3. In the Introduction section, the author needs to provide or add an explanation or review of some methods that can be done to overcome environmental problems, especially dye removal. This is because there are still many methods that can be done to overcome the environmental problems of dye removal. This is also necessary so that readers can find out the reasons or considerations why in this study the author prefers to use absorption over some other methods.

4. Introduction section is not convincing. A more in-depth current literature survey needs to be added in the introduction section.

5. Rewrite this section again completely.

6. On page 1 authors have mentioned “The adsorption method was widely regarded as the most effective way 38 to treat dye wastewater because of its significant adsorption capacity, low cost,  good selectivity, and easy operation”. Kindly add the following relevant references in support of the sentences. International Journal of Biological Macromolecules 223 (A), 636-651, 2022; Environmental Research 214, 114000, 2022; Water 14 (21), 3411, 2022.

7. The author needs to provide or add a sub-section "Chemicals" which contains detailed information about all chemicals used in this study (chemical formulas, purity, and other information).

8. The author needs to add and determine the pH of zero point charge (pHzpc) for CaO-C3N4 nanosorbent.

9. Keywords: avoid using molecular formula.

10. More discussion should be presented after the characterization in section 3.1

11. On page 8, Figure 5. (a) Effect of pH on % removal of BF, (b) Influence of dye concentration onto the adsorption capacity  of CaO-C3N4 nanosorbent. Provide the error bar in the graphs.

12. On page 10, Figure 7. (a) Equilibrium time study (B) PFO, (b) PSO and (d) Intra-particle diffusion graphs for the uptake 211 of BF onto CaO-C3N4. The results shown in the graphs are not convincing. The authors need to repeat this experiment and carefully correctly calculate the kinetics adsorption models.

13. On page 12, section 3.3. Regeneration and reusability study, how the author performed the experiments. Provide the stepwise process. And how the desorption of BF dye was done after every cycle.

14. Section 3.5. Adsorption mechanism needs more in-depth and clear discussion.

Author Response

  1. Title of the script is not convincing. Need to be revised.

Reply:

The title was revised.

  1. The author needs to confirm and clarify the novelty of this research.

Reply:

A novel CaO-g-C3N4 nanosorbent with high surface area and the excellent adsorption capability ‎was successfully fabricated by simple ultrasonic method. The adsorption performance of dye selected as pollutants following the pseudo-second-order kinetics and Langmuir model due to chemosorption process.

  1. In the Introduction section, the author needs to provide or add an explanation or review of some methods that can be done to overcome environmental problems, especially dye removal. This is because there are still many methods that can be done to overcome the environmental problems of dye removal. This is also necessary so that readers can find out the reasons or considerations why in this study the author prefers to use absorption over some other methods.

  Reply:

More an explanation and review of some methods were added to introduction section as per the reviewer's request.

  1. Introduction section is not convincing. A more in-depth current literature survey needs to be added in the introduction section.

Reply:    

 We agree with the reviewer’s point and the manuscript has improved and more recent information’s were added as per the reviewer's request.

  1. Rewrite this section again completely.

Reply:

The introduction section was improved and rewritten as per the reviewer's request.

  1. On page 1 authors have mentioned “The adsorption method was widely regarded as the most effective way 38 to treat dye wastewater because of its significant adsorption capacity, low cost, good selectivity, and easy operation”. Kindly add the following relevant references in support of the sentences. International Journal of Biological Macromolecules 223 (A), 636-651, 2022; Environmental Research 214, 114000, 2022; Water 14 (21), 3411, 2022.

Reply:

The relevant references were cited as per the reviewer's request.

  1. The author needs to provide or add a sub-section "Chemicals" which contains detailed information about all chemicals used in this study (chemical formulas, purity, and other information).

Reply:

The authors acknowledge the reviewer for the valuable recommendation and the chemical formulas and the grades of all used precursors have been added in the experimental section.

  1. The author needs to add and determine the pH of zero point charge (pHzpc) for CaO-C3N4 nanosorbent.

The pH of zero point charge (pHzpc) for CaO-C3N4 nanosorbent was added (Fig5b).

  1.  Keywords: avoid using molecular formula.

Reply:

The authors thank the Reviewer for this remark. The molecular formula was avoided.

  1. More discussion should be presented after the characterization in section 3.1

Reply:

More information’s and new characterizations results such as XPS, pore distributions and TEM images of CaO nanomaterials and pure g-C3N4 nanosheets had been added and discussed as per the reviewer's request.

  1. On page 8, Figure 5. (a) Effect of pH on % removal of BF, (b) Influence of dye concentration onto the adsorption capacity  of CaO-C3N4 nanosorbent. Provide the error bar in the graphs.

Reply:

Thank you for your keen observation. A error bar ‎ has been added.

  1. On page 10, Figure 7. (a) Equilibrium time study (B) PFO, (b) PSO and (d) Intra-particle diffusion graphs for the uptake 211 of BF onto CaO-C3N4. The results shown in the graphs are not convincing. The authors need to repeat this experiment and carefully correctly calculate the kinetics adsorption models.

Reply:

The authors acknowledge the reviewer for the valuable recommendation and the kinetics study was evaluated using nonlinear pseudo-first-order (PFO), and pseudo-second-order (PSO) models.

  1. On page 12, section 3.3. Regeneration and reusability study, how the author performed the experiments. Provide the stepwise process. And how the desorption of BF dye was done after every cycle.

Reply:

The authors apologize for the missing of this information. The stepwise process of regeneration and reusability study have been now added in the experimental section.

" For the reusability test, the CaO-g-C3N4 nanosorbent used after the adsorption experiments was recovered by filtration and then calcined at 500° C. for one hour. After that, the recovered CaO-g-C3N4 nanosorbent was reused for further adsorption tests."

  1. Section 3.5. Adsorption mechanism needs more in-depth and clear discussion.

Reply:

More discussion was added to clarify the adsorption mechanism.

Reviewer 3 Report

In my opinion, the article needs to be refined and provide more details.

1) Introduction should be developed. For example, there is no information about the purpose of the research. Please also write why g-C3N4 is extensively studied.

2) In experimental please provide more details:

- names of reagent suppliers

- names of equipment manufacturers

- “By mixing 25 mL of BF dye solution with 10 mg of CaO-g-C3N4 nanosorbent at varying starting concentrations (5-300 mg/L), batch removal tests of BF dye were conducted.” please provide the used initial concentrations (not only range).

- “Later, 10 ml of the suspension was withdrawn and centrifuged at certain time intervals to measure the remaining concentration of BF dye.” Please provide the value of time intervals.

3) Results and discussions

- Authors wrote” Due to the presence of several active sites on the surface, the CaO-g-C3N4 nanosorbent's increased surface area, demonstrated by a higher specific surface area, will improve the sorption capacity [25]” What was the size of the specific surface area and pore size?

-Regeneration and reusability study – please provide in the text how much the efficiency of BF removal decreased after each cycle.

- Fig. 8b, please mark the mentioned in the text bands on the FTIR spectra. 

Author Response

1) Introduction should be developed. For example, there is no information about the purpose of the research. Please also write why g-C3N4 is extensively studied.

The introduction section was improved and rewritten as per the reviewer's request.

2) In experimental please provide more details:

- names of reagent suppliers

- names of equipment manufacturers

The authors acknowledge the reviewer for the valuable recommendation and, more details have been added in the experimental section.

- “By mixing 25 mL of BF dye solution with 10 mg of CaO-g-C3N4 nanosorbent at varying starting concentrations (5-300 mg/L), batch removal tests of BF dye were conducted.” please provide the used initial concentrations (not only range).

The used initial concentrations were added.

- “Later, 10 ml of the suspension was withdrawn and centrifuged at certain time intervals to measure the remaining concentration of BF dye.” Please provide the value of time intervals.

The values of intervals time (10 min) was provided.

3) Results and discussions

- Authors wrote” Due to the presence of several active sites on the surface, the CaO-g-C3N4 nanosorbent's increased surface area, demonstrated by a higher specific surface area, will improve the sorption capacity [25]” What was the size of the specific surface area and pore size?

The specific surface area and pore volume were added.

-Regeneration and reusability study – please provide in the text how much the efficiency of BF removal decreased after each cycle.

The efficiency of BF removal was provided in the text.

- Fig. 8b, please mark the mentioned in the text bands on the FTIR spectra. 

The mentioned FTIR bands was marked on the FTIR spectra. 

Round 2

Reviewer 1 Report

The manuscript can be accepted in present form.

Author Response

RESPONSE TO EDITOR COMMENTS

Title: “Uptake of BF Dye from the aqueous phase by CaO-g-C3N4 nanosorbent: construction, descriptions, and recyclability,”

Manuscript ID: inorganics-2046515

The authors are very thankful to the editor for his useful comments and suggestions. All comments made by the editor have been carefully considered, and the manuscript has been modified to meet the standards of the Inorganics Journal. The corrections incorporated in the revised manuscript are highlighted in red.

Response to Editor Comments

  • The revised version of the title is not consistent. This Editor recommends to propose a new one.

Response:

This paper comes within the framework of a research project supported by our university. Among the conditions for accepting funding, the research title must be fixed and therefore cannot be changed. Thank you for  understanding.

  • Correspondence: all authors should add their email address (not only the corresponding one). By the way, all email addresses should contain the name of the author and not just numbers. This way it is possible to certify that the manuscript is written unequivocally by the authors.

Response:

Correspondence and all authors were added with their email address as editor request.

  • Abstract should be revised in a proper manner reflecting the content of the manuscript.

Response:

The abstract was revised and improved as editor request.

  • 8. Authors cited FTIR data but they did not show/discuss these data, the signals attribution and so on.

Response:

The mistake was corrected as editor request.

  • FTIR spectra reported in Figure 8A are very poor in terms of quality. In particular the signals are very close to the saturation. These spectra should be completely redone.

Response:

FTIR spectra were enhanced as editor request.

  • Table 3. Authors compared their performance with other case studies from the literature. This Editor finds this choice not properly correct as the comparison should be performed over the entire class of basic dye, reporting the experimental conditions. This way the comparison will evidence also case studies with samples howling higher sorption capacities. Furthermore, it seems quite curious that authors do not report the performance of MgO/g-C3N4 recently published by themselves (DOI: 10.1016/j.envres.2021.112543) where superior sorption performances where reached.

Response:

We are very grateful for the editor insightful comment. Table 3 modified as editor request.

  • What is really surprising is that authors do not perform sorption tests with bare g-C3N4 as it seems that the mechanism requires only the presence of g-C3N4. This is mandatory.

Response:

Thanks for appreciated comment, the sorption of BF dye onto bare g-C3N4 was tested and the obtained results was added in Fig 6c and discussed in the manuscript.

  • In the entire manuscript it is not clear why authors included CaO? What is the role of CaO?

Response:

By mixing g-C3N4 nanosheets with metal oxides, several ways have been applied to increase their surface activity. The recommendation is to select CaO nonmaterial due to the strong basicity of CaO and its application as an adsorbent for the removal of toxic waste cleanup. Calcium oxide is also an effective adsorbent for detoxifying dye-contaminated water discharged by textile companies and pharmaceutical and agricultural wastes at a cheap cost.

  • This Editor would like authors to highlight what is the novelty of their paper compared to the literature.

Response:

A mesoporous CaO@g-C3N4 nanocomposite was successfully produced using a simple sonochemical process and evaluated as a promising adsorbent material for adsorbing the basic fuchsin dye from a contaminated aqueous phase.

  • Lastly, in the Introduction authors wrote "Nevertheless, sorption isotherms have been extensively studied in relation to the use of nanomaterials with high surface areas, such as metal oxide doped g-C3N4, in wastewater treatment systems; however, no studies have looked at the behavior of sorption at solid/liquid interfaces". This is not realistic; the literature is plenty of studies enlighting the adsorbent-adsorbate interaction (some of them even written by this Editor). Therefore, this Editor strongly recommends authors to deeply analyze the scientific literature.

Response:

We are very grateful for the editor insightful comment. The introduction was analyzed and modified according to scientific literature as editor request.

Reviewer 2 Report

The authors have satisfactorily revised the manuscript.

Author Response

(The authors gave the same response as above.)

Reviewer 3 Report

The authors made the necessary corrections. In my opinion, the article is suitable for publication in its current form.

Author Response

(The authors gave the same response as above.)
